# A Novel Planar Power Divider/Combiner for Wideband High-Power Applications

Hamed Tadayon [1], Mansoor Dashti Ardakani [1,*], Reza Karimian [2], Shahrokh Ahmadi [2] and Mona Zaghloul [2]

[1] Institut National de la Recherche Scientifique (INRS-EMT), University of Quebec, Montreal, QC H5A 1K6, Canada
[2] Department of Electrical Engineering, George Washington University (GWU), Washington, DC 20052, USA
* Correspondence: mansoor.dashti@gmail.com

**Abstract:** This manuscript presents a novel wideband power divider with high power handling capability. The power handling capacity of the power divider is increased through the use of grounded 50 ohm loads. The full circuit analysis of a single section of the proposed structure is presented utilizing even and odd modes and ABCD matrices. The final designed sections are cascaded and extended to achieve a high bandwidth response for the target of X-band. The structure was designed and optimized with the method of moments based on ADS software and simulated in HFSS for 3D full-wave analysis. A prototype module was fabricated and measured for experimental validation. The simulation results were confirmed by through measurements for the frequency band of 7–12 GHz (more than 52% fractional bandwidth). A divider such as the proposed one has a significantly higher power handling capacity than Wilkinson, as well as a wider frequency bandwidth than Gysel power dividers. As a wideband high-power power divider, the proposed device is ideal for high-data rate radar or satellite applications.

**Keywords:** ABCD matrix; grounded resistors; Wilkinson power splitter; Gysel power divider; high power handling; wideband applications; telecommunications





## 1. Introduction

Power dividers have many applications in telecommunication systems, such as power amplifiers and array antennas [1–3]. The Wilkinson power divider is a popular component in radio frequency and microwave circuits, and it has a critical role in several technologies, systems, and configurations, such as array antenna feeding networks [4–6], multi-port technologies/sensors [7–9], and non-reciprocal systems [10–12]. Dual-band Wilkinson power dividers have been developed using different ways in recent years, along with the introduction of the multiband wireless communication system [13–15]. These dual-band Wilkinson power dividers can only be terminated with nominal impedances. Many microwave components, such as transistors, diodes, and antennas, cannot be utilized directly because their input or output ports have a frequency-dependent complex impedance [16]. To accomplish impedance matching, an additional impedance transformer must be placed between Wilkinson power dividers and microwave components. As a result, in practice, broadband Wilkinson dividers require resistors with different values and cannot handle high powers [17].

On the other hand, Gysel power dividers (GPDs) have a higher power handling capability [18]. In comparison with Wilkinson power dividers, Gysel power dividers' use of grounded resistors is one of their advantages [19]. GPDs can handle more power by using grounded resistors, which transfer heat to the ground. A number of works have been published on Gysel power dividers [20–23], but these designs have a low ratio of frequency bandwidth. Figure 1 shows the structure of conventional Wilkinson and Gysel Power dividers. To achieve a greater bandwidth, the multi-section model cannot be implemented

in GPDs due to their specific structure. So, we are limited to designing a GPD at a central frequency with a fixed bandwidth [18]. To the knowledge of the authors, there has not been any reported paper on the bandwidth enhancement of a GPD.

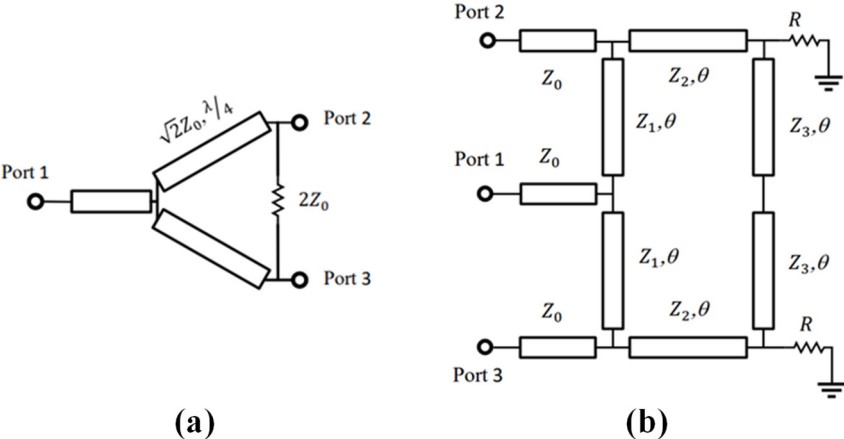

**Figure 1.** Conventional power dividers: (**a**) Wilkinson power divider and (**b**) Gysel power divider.

In this paper, a novel wideband power divider with high power handling capability is proposed. A divider such as the proposed one has a significantly higher power handling capacity than Wilkinson, as well as a wider frequency bandwidth than GPD. With multiple sections stacked together, a wider bandwidth can be achieved at the output ports of the final structure. The multi-section Wilkinson configuration requires several resistors with different values [24], whereas the proposed multi-section power divider requires only grounded resistors for isolation purposes, and the resistor's value is equivalent to the design's characteristic impedance.

In order to understand the theory of the proposed power divider, the circuit analysis of a single section has been provided. The designed power divider is operating over the entire X-band frequency (i.e., 7–12 GHz). For verification, a prototype of the HFSS-modeled design was implemented and measured with a vector network analyzer (VNA). As reported herein, the simulation and measurement results were found to be in excellent agreement.

## 2. Materials and Methods

### 2.1. Theoritical Analysis

Figure 2 shows the equivalent circuit model of the proposed power divider for one section of the structure. The power reaches ports 2 and 3 with equal phase and amplitude in the divider mode. All transmission lines have the same length as a quarter wavelength. The proposed structure can be solved using even and odd analysis. Figure 3a depicts the even model and Figure 3b shows the odd model of the structure when ports 2 and 3 are excited. Figure 4a illustrates a half-circuit of the even mode that can be considered as a combination of two other circuits, which are shown in Figure 4b,c. Equations can be solved using ABCD matrix relations.

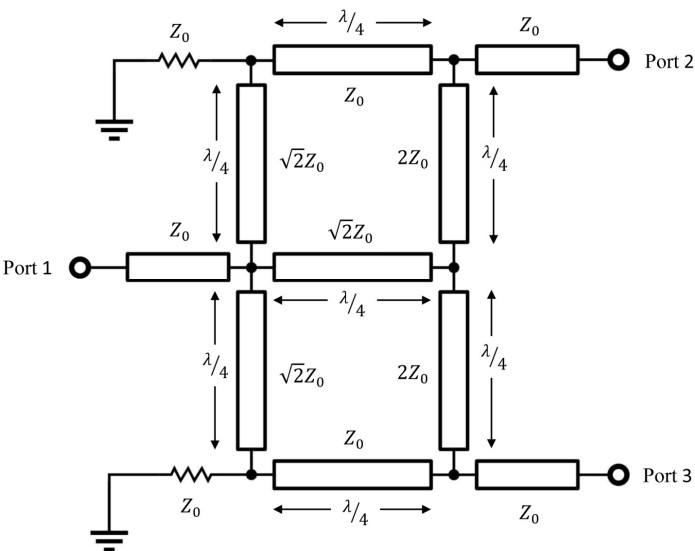

**Figure 2.** The equivalent circuit model of one section of the proposed power divider.

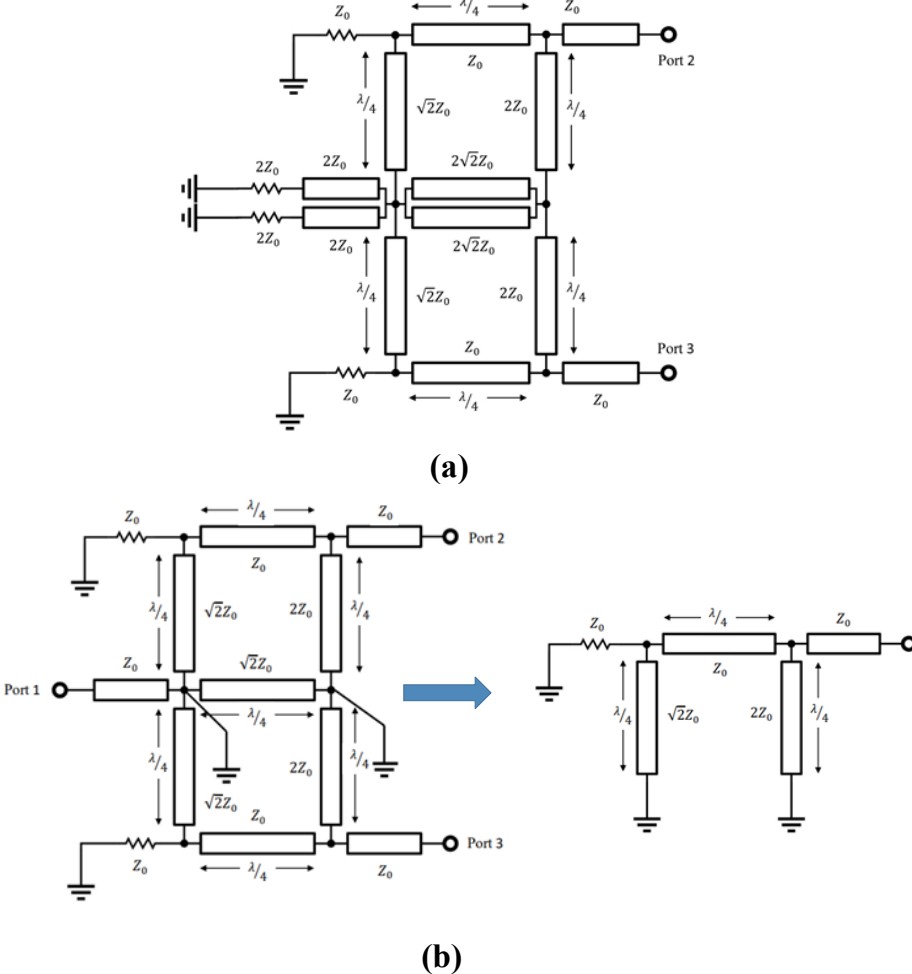

**Figure 3.** (**a**) Even mode and (**b**) odd mode equivalent circuits of the proposed power divider.

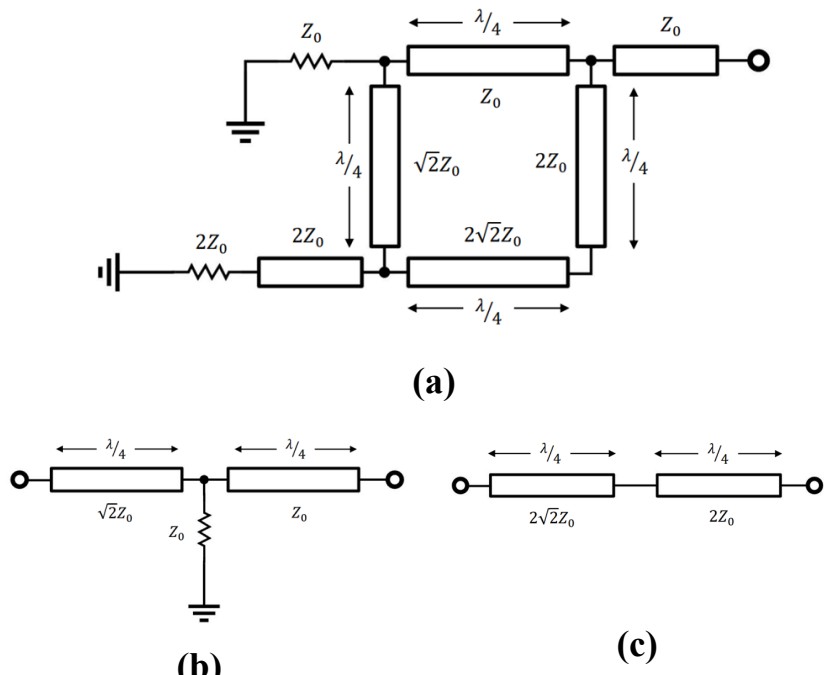

**Figure 4.** The half-circuit of the even mode equivalent circuits of the proposed power divider, simplified into three separate circuits (**a**) overall circuit (**b**) equivalent T-network (**c**) equivalent series network.

The ABCD matrix of the structure in Figure 4b is as follows:

$$\text{ABCD} = \begin{bmatrix} 0 & j\sqrt{2}Z_0 \\ \frac{j}{\sqrt{2}Z_0} & 0 \end{bmatrix} \begin{bmatrix} 1 & 0 \\ \frac{1}{Z_0} & 1 \end{bmatrix} \begin{bmatrix} 0 & jZ_0 \\ \frac{j}{Z_0} & 0 \end{bmatrix} = \begin{bmatrix} -\sqrt{2} & -\sqrt{2}Z_0 \\ 0 & -\frac{1}{\sqrt{2}} \end{bmatrix}$$

and the ABCD matrix of the structure in Figure 4c is as follows:

$$\text{ABCD} = \begin{bmatrix} 0 & j2\sqrt{2}Z_0 \\ \frac{j}{2\sqrt{2}Z_0} & 0 \end{bmatrix} \begin{bmatrix} 0 & j2Z_0 \\ \frac{j}{2Z_0} & 0 \end{bmatrix} = \begin{bmatrix} -\sqrt{2} & 0 \\ 0 & -\frac{1}{\sqrt{2}} \end{bmatrix}$$

On the basis of Figure 5, considering two parallel networks of even mode half-circuits, we have the following results for the full circuit:

$$I_2 = I_{12} + I_{22}$$

$$\begin{bmatrix} V_1 \\ I_{21} \end{bmatrix} = \begin{bmatrix} -\sqrt{2} & 0 \\ 0 & -\frac{1}{\sqrt{2}} \end{bmatrix} \begin{bmatrix} V_2 \\ -I_{22} \end{bmatrix} \Rightarrow \left\{ \begin{array}{l} I_{21} = \frac{1}{\sqrt{2}}I_{22} \\ V_1 = -\sqrt{2}V_2 \end{array} \right\}$$

$$\begin{bmatrix} V_1 \\ I_{11} \end{bmatrix} = \begin{bmatrix} -\sqrt{2} & -\sqrt{2}Z_0 \\ 0 & -\frac{1}{\sqrt{2}} \end{bmatrix} \begin{bmatrix} V_2 \\ -I_{12} \end{bmatrix} \Rightarrow I_{11} = \frac{1}{\sqrt{2}}I_{12}$$

$$I_1 - I_{21} = \frac{1}{\sqrt{2}}(I_2 - I_{22}) \Rightarrow I_1 = \frac{1}{\sqrt{2}}I_2 + I_{21} - \frac{1}{\sqrt{2}}I_{22} \overset{3}{\Rightarrow} I_1 = \frac{1}{\sqrt{2}}I_2$$

$$\Rightarrow \begin{bmatrix} V_1 \\ I_1 \end{bmatrix} = \begin{bmatrix} -\sqrt{2} & 0 \\ 0 & -\frac{1}{\sqrt{2}} \end{bmatrix} \begin{bmatrix} V_2 \\ -I_2 \end{bmatrix}$$

$$\Rightarrow \begin{bmatrix} V_2 \\ I_2 \end{bmatrix} = \begin{bmatrix} -\frac{1}{\sqrt{2}} & 0 \\ 0 & -\sqrt{2} \end{bmatrix} \begin{bmatrix} V_1 \\ -I_1 \end{bmatrix}$$

$$\Rightarrow \begin{bmatrix} V_2 \\ I_2 \end{bmatrix} = \begin{bmatrix} -\dfrac{1}{\sqrt{2}} & 0 \\ 0 & -\sqrt{2} \end{bmatrix} \begin{bmatrix} V_1 \\ -I_1 \end{bmatrix}$$

**Figure 5.** Block diagram of two parallel networks.

The last relation shows the ABCD matrix of the even mode, and according to the input impedance relation for Figure 4a, we have

$$Z_{in} = \frac{AZ_L + B}{CZ_L + D} \quad \overset{Z_L = 100\ \Omega}{\Longrightarrow} \quad Z_{in} = \frac{-\frac{1}{\sqrt{2}} \times 100}{-\sqrt{2}} = 50\ \Omega$$

As a result, the circuit is matched in the even mode. For the odd mode, the circuit is simplified as shown in Figure 3b, which is fully matched due to the open circuit of the quarter wavelength transmission line.

### 2.2. Modeling

The schematic of the power divider was modeled in ADS software and cascaded up to four sections to provide the target bandwidth. The parametric schematic was optimized with the method of moments based on the ADS Momentum tool. The final ADS schematic for this design is illustrated in Figure 6. The final layout was imported to HFSS software for a 3D full-wave analysis and more accurate results. In HFSS, a wave port is commonly used for excitation, and the software generates a solution by exciting each wave port individually. Each mode incident on a port contains one watt of time-averaged power. Another important feature is the definition of the radiation boundary. Since this design is an open boundary problem, radiation boundaries are used to emulate a wave radiating infinitely far into space.

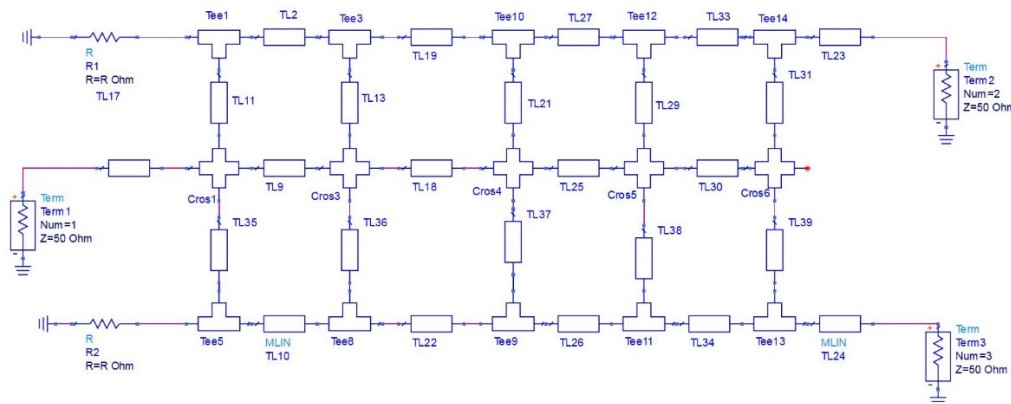

**Figure 6.** A parametric subnetwork schematic design model of the power divider in ADS.

### 3. Results and Discussion

As mentioned, a circuit analysis was performed in ADS software, while a full-wave analysis was carried out in HFSS. Initially, a single section of the structure was designed,

which was extended to a multi-section structure for a wider bandwidth in the 7–12 GHz frequency range. Figure 7 shows the final multi-section structure of the proposed power divider. The optimized dimensions are also depicted in Figure 7. The presented structure has been manufactured on a substrate in order to validate its performance. Rogers RT5880 with a dielectric permittivity of 2.2, dielectric thickness of 0.508 mm, copper thickness of 35 μm, and loss tangent of 0.0005 has been utilized as the dielectric substrate. The module has been fabricated with the standard printing process for the microstrip components. In addition, for measurement purposes, all five ports are connected to 50 ohm SMA connectors that operate up to 18 GHz. Isolated ports are terminated with available low-power SMA termination loads; however, it should be noted that in high-power applications, high-power 50 ohm resistors should be used.

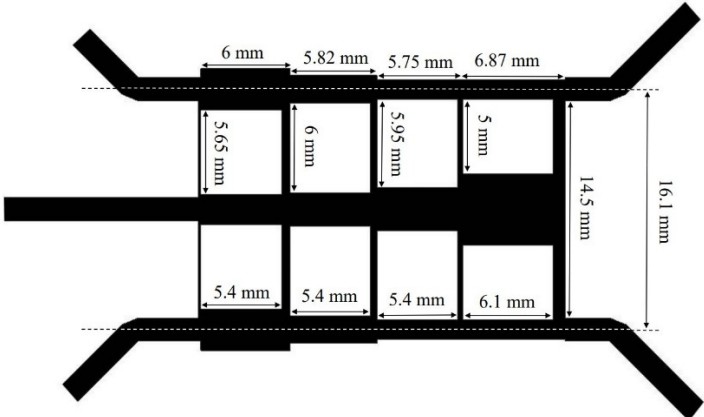

**Figure 7.** Prototype of the final designed power divider and its dimensions.

Figure 8 is a photo of the fabricated power divider. The simulation and measurement results for the insertion loss and also the isolation between output ports are shown in Figure 9. For power dividers, the insertion loss is defined by the additional loss above the nominal loss due to splitting. For the proposed 3 dB power divider, the average obtained insertion loss ($S_{21}$) is about 0.5 dB. The isolation between output ports ($S_{32}$) is better than −15 dB. This means that a signal entering output 2 does not leak out of output 3, which is essential for most applications, such as diplexers in MIMO systems [25].

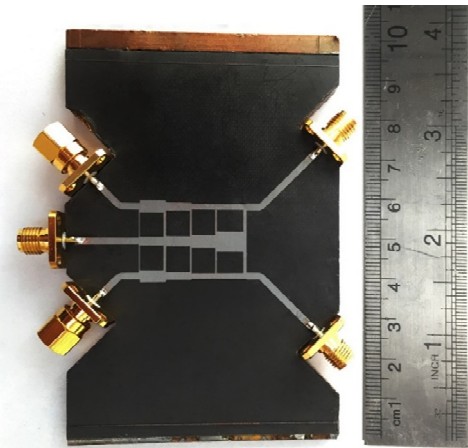

**Figure 8.** A photo of the fabricated power divider module.

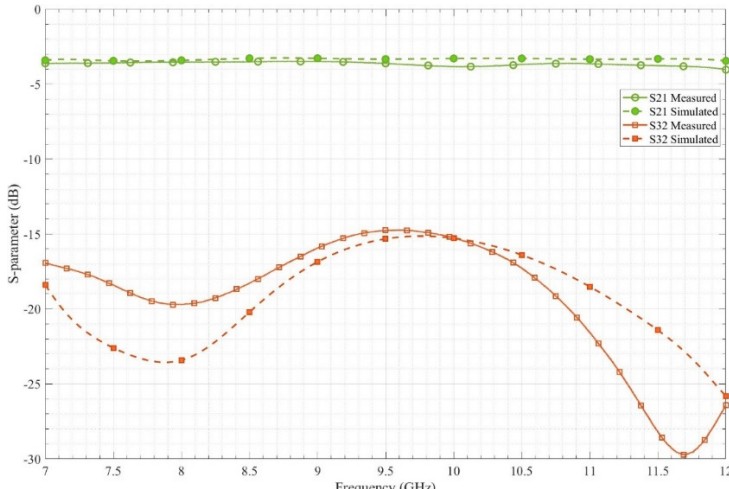

**Figure 9.** The achieved simulation and measurement results of insertion loss and output ports' isolation for the proposed power divider.

Figure 10 presents the return losses for both input and output ports. The input port (Port 1) has a perfect matching (better than −16 dB), and this value is lower than −11 dB for both output ports. If the input return loss is kept below −13 dB, the transmitted signal will not be affected by reflected signals. As one can see, there is good agreement between the simulation and measurement results.

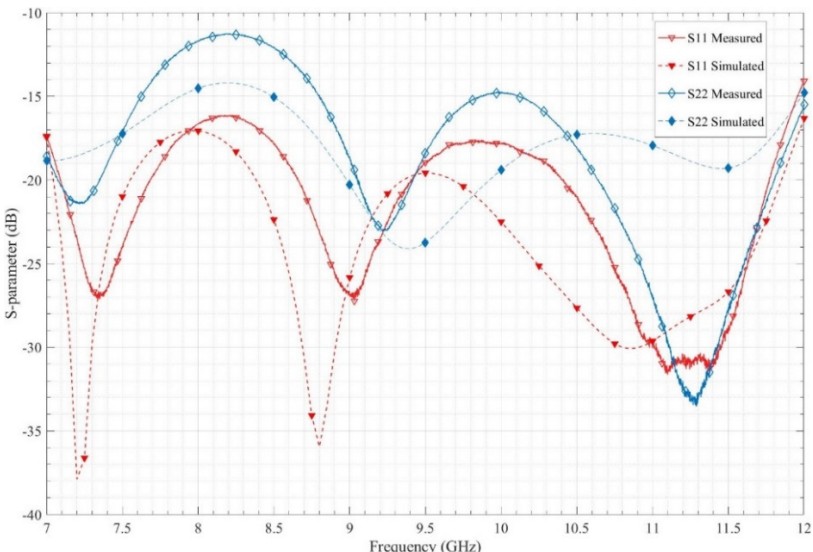

**Figure 10.** The simulated and measured return losses for the designed power divider.

Nonetheless, there are some differences between the simulation and experimental results. It is important to note that the SMA connectors are soldered to the implemented module and they are not simulated. That is the primary cause of these differences, especially for insertion and return losses. In addition, we have measured the isolation parameter ($S_{32}$) using the available SMA termination loads as grounded resistors. It is reasonable to expect that these loads do not behave like ideal ones at such a high frequency. Finally, it should be noted that experimental results are always subject to measurement and calibration errors.

## 4. Conclusions

In this paper, a novel power divider is presented for both wideband and high-power applications. Cascading several sections of this power divider increases the operating frequency bandwidth for broadband performance. This design also takes advantage

of grounded resistors to transfer heat to the ground, which allows it to handle more power. The proposed power divider has been analyzed, modeled, simulated, fabricated, and measured. The measured results of the prototype are in good agreement with the simulation and validate the idea. In both simulations and measurements, the designed power divider proved to have better than $-15$ dB of return losses and low insertion losses over a 52% fractional bandwidth (from 7 to 12 GHz). In comparison with similar works, the proposed power divider has demonstrated its high power handling and capability to simultaneously provide wide bandwidth and good isolation, making it an excellent candidate for broadband high-power applications such as satellites and radars.

**Author Contributions:** Conceptualization, H.T. and M.D.A.; methodology, H.T., R.K. and M.D.A.; validation, M.D.A.; formal analysis, H.T.; investigation, H.T.; resources, S.A. and M.Z.; data curation, S.A.; writing—original draft preparation, R.K.; writing—review and editing, M.D.A.; visualization, M.D.A.; supervision, M.Z.; project administration, S.A.; funding acquisition, M.Z. All authors have read and agreed to the published version of the manuscript.

**Funding:** This research received no external funding.

**Data Availability Statement:** Not applicable.

**Conflicts of Interest:** The authors declare no conflict of interest.

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
