# Peer review of "A Novel Planar Power Divider/Combiner for Wideband High-Power Applications"

_2673-4117, doi:10.3390/eng3040033_

Round 1

Reviewer 1 Report

Authors have presented a study on A Novel Planar Power Divider/Combiner for Wideband High Power Applications. However, I will not recommend this paper in the current form because it has short some shortcoming. In order to improve this paper, my comments are given below.

1.       Abstract is too general, no novelty is discussed. Quantitative results are missing.

2.       There is no sufficient literature. add more literature and identify the research gap.

3.       Objective of the paper are missing.

4.       Results are not sufficient.

5.       Mythology is not given.

6.       In conclusion, quantitative results are missing.

Author Response

Dear reviewer,

Thank you for your comments and feedback regarding our manuscript. In response to all the reviewer's comments, all of the raised concerns have been addressed in the uploaded file. A version with the highlighted changes is included so that the modifications will be more visible. Please see the attachments. 

Reviewer 2 Report

Dear Authors,

Please find my guidelines in the attachment. 

Kind Regards

Author Response

Dear reviewer,

Thank you for your comments and feedback regarding our manuscript. we would like to manifest our appreciation for having brought important point of views to our attention during the revision of the paper.

In response to the reviewer's comments, all of the raised concerns have been addressed point-by-point in the uploaded file. A version with the highlighted changes is included so that the modifications will be more visible. Please see the attachments. 

Round 2

Reviewer 1 Report

Now paper can be accepted

Author Response

We would like to thank you again for considering our paper and taking the time to read and review it.

Reviewer 2 Report

Dear Authors,

Thank you for taking into consideration my suggestions. I appreciate it. I only have one more comment - the references list still includes about 70% of self-citations. Please change it. 

Thus, I suggest the MINOR revision. 

Kind regards

Author Response

Thank you again for the time and consideration. For references, we cited the relevant references that we were aware of their contents. This design is the continuation of the authors' previous publications, so readers should refer to our previous publications. Anyhow, the reviewer's comments have been respected, and some references have been replaced with more relevant ones. Please see the attached highlighted file.
